# Development of a One-Step Multiplex qPCR Assay for Detection of Methicillin and Vancomycin Drug Resistance Genes in Antibiotic-Resistant Bacteria

**DOI:** 10.3390/pathogens13100853

**Published:** 2024-09-30

**Authors:** Jiyoung Lee, Eunyoung Baek, Hyesun Ahn, Jinyoung Bae, Sangha Kim, Sohyeong Kim, Suchan Lee, Sunghyun Kim

**Affiliations:** 1Department of Research & Development, DreamDX Inc., C001, 57, Oryundae-ro, Geumjeong-gu, Busan 46252, Republic of Korea; lab84@dreamdx.net (J.L.); eybaek@dreamdx.net (E.B.); 2Joint & Arthritis Research Center, Himchan Hospital, 120, Sinmok-ro, Yangcheon-gu, Seoul 07999, Republic of Korea; ahs0614@naver.com; 3Department of Nano-Bio Convergence Division, Korea Institute of Materials Science, 797 Changwondae-ro, Changwon 51508, Republic of Korea; bjyscene@kims.re.kr; 4Department of Laboratory Medicine, Konyang University Hospital, 158, Gwanjeodong-ro, Seo-gu, Daejeon 35365, Republic of Korea; comi0110@hanmail.net; 5Department of Clinical Laboratory Science, College of Health Sciences, Catholic University of Pusan, Busan 46252, Republic of Korea; hyeongso29@naver.com; 6Next-Generation Industrial Field-Based Specialist Program for Molecular Diagnostics, Brain Busan 21 Plus Project, Graduate School, Catholic University of Pusan, Busan 46252, Republic of Korea

**Keywords:** methicillin-resistant *Staphylococcus aureus*, vancomycin-resistant enterococci, quantitative PCR, molecular diagnosis, resistance gene

## Abstract

The most common antibiotic-resistant bacteria in Korea are methicillin-resistant *Staphylococcus aureus* (MRSA) and vancomycin-resistant enterococci (VRE). Pathogen identification in clinical laboratories can be divided into traditional phenotype- and genotype-based methods, both of which are complementary to each other. The genotype-based method using multiplex real-time polymerase chain reaction (PCR) is a rapid and accurate technique that analyzes material at the genetic level by targeting genes simultaneously. Accordingly, we aimed to develop a rapid method for studying the genetic characteristics of antibiotic-resistant bacteria and to provide an experimental guide for the efficient antibiotic resistance gene analysis of *mecA* detection for MRSA and *vanA* or *vanB* detection for VRE using a one-step multiplex qPCR assay at an early stage of infection. As a result, the sensitivity and specificity of the *mecA* gene for clinical *S. aureus* isolates, including MRSA and methicillin-susceptible *S. aureus*, were 97.44% (95% CI, 86.82–99.87%) and 96.15% (95% CI, 87.02–99.32%), respectively. The receiver operating characteristic area under the curve for the diagnosis of MRSA was 0.9798 (*** *p* < 0.0001). Therefore, the molecular diagnostic method using this newly developed one-step multiplex qPCR assay can provide accurate and rapid results for the treatment of patients with MRSA and VRE infections.

## 1. Introduction

Antibiotic resistance was considered a serious threat to global public health by the WHO and applies to both developed and underdeveloped countries [1,2]. Higher overall resistance rates among major bacteria were detected in Asian countries including Korea relative to those of Western countries [2,3,4]. The causative agents of the six types of statutory infectious diseases in South Korea are methicillin-resistant *Staphylococcus aureus* (MRSA), vancomycin-resistant enterococci (VRE), vancomycin-resistant *S. aureus* (VRSA), multidrug-resistant *Pseudomonas aeruginosa* (MRPA), multidrug-resistant *Acinetobacter baumannii* (MRAB), and carbapenem-resistant Enterobacteriaceae (CRE) groups of bacteria [5,6]. In South Korea, several studies have reported that MRSA and VRE infections are the highest among the six major antibiotic-resistant bacterial infections [7,8,9,10].

Currently, clinical laboratories use several AST methods, including disk diffusion and minimum inhibitory concentration (MIC), depending on the equipment and their laboratory test manuals [11,12]. Classical culture-dependent methods have been firmly established in the diagnostic routine, with the main limitation being that results for the most clinically important bacteria are obtained in at least 18–24 or 48 h [12,13]. The diameter should be measured visually or using an automated system, and the results obtained should be interpreted and classified according to the recommended clinical guidelines of the standard in use [14].

Additionally, as an auxiliary molecular diagnostic technology for identifying antibiotic-resistant bacteria, a genotype-based method using real-time polymerase chain rection (PCR) was applied, and this analysis was performed by targeting the *mecA*, *vanA*, or *vanB* genes of MRSA and VRE, respectively [15,16,17,18]. According to these guidelines, MRSA diagnosis is based on the confirmation of oxacillin or cefoxitin antibiotic resistance in *S. aureus* isolates from clinical specimens and the detection of the methicillin resistance-specific gene (*mecA*). In addition, VRE is based on the confirmation of vancomycin antibiotic resistance in enterococci isolated from clinical specimens and the detection of vancomycin resistance-specific genes (*vanA* or *vanB*) [18,19,20].

The continued emergence of resistance to currently available antimicrobial agents highlights the importance of preventing the transmission and infection of these microorganisms [21]. MRSA and VRE, major hospital infectious strains, have been reported to cause resistance prevalence through the horizontal transfer of resistance genes [22,23,24,25]. MRSA acquires resistance to beta-lactam antibiotics via the *mecA* gene [26], which is located on the chromosomal mobile genetic element staphylococcal cassette chromosome *mec* (SCC*mec*) [27] and is 2.1 kb long [28]. Among the clinically isolated *S. aureus*, the *mecA* gene was mainly detected in the MRSA isolates [29]. In South Korea, hospital isolates of VRE are reported to have vancomycin-resistant characteristics by possessing the *vanA* and *vanB* genes [25,30,31], of which resistance transmission has been demonstrated [32,33]. In VRE, nine resistance gene types, *vanA*, *vanB*, *vanC*, *vanD*, *vanE*, *vanG*, *vanL*, *vanM*, and *vanN*, which are resistant to glycopeptide antibiotics, have been identified and divided into nine resistance types according to the base sequences of the resistance ligases [25,31,33,34,35]. Among several types, *vanA* and *vanB* were the predominant vancomycin resistance gene clusters associated with transposon sequences Tn1546 and Tn1549, respectively [36].

Currently, microbial culture and phenotypic AST play important roles in diagnosis. However, phenotypic AST can be time-consuming, and this is the main limitation of this method, owing to the need for microbial growth. The lag phase is the early stage of microbial growth and a time-limiting factor. A variety of spectral methods are available to enhance the speed of results, but sophisticated microscopy is required for visualizing the cells. Additionally, the appropriate detection of heterogeneity is clinically important but currently difficult with AST. The development of techniques for the potential detection of heterogeneity is crucial, as individual cells within a bacterial population may have different antimicrobial susceptibilities or exhibit different levels of resistance [37]; therefore, it is necessary to fully promote molecular diagnostics to compensate for these shortcomings and enable rapid diagnosis. Many molecular diagnostic tests have been developed to overcome the limitations of microbiological tests, and research and the commercialization of diagnostic kits are currently underway. For rapid diagnosis, a genotypic method that improves the time and cost aspects of the current PCR methods used is needed. Moreover, developing the simultaneous diagnostic detection of MRSA and VRE in this study would reduce processing time, increase specimen throughput [38], and overcome these limitations through a one-step multiplex qPCR assay. In addition, MRSA and VRE, which account for most of the antibiotic resistance in South Korea, have a high multidrug resistance (MDR) rate [39]; thus, the detection efficiency can be increased by a one-step multiplex qPCR assay.

Therefore, the aim of this study is to design a technique to simultaneously detect *mecA* and *vanA* or *vanB* genes specific for MRSA and VRE. We aim to develop a novel one-step multiplex qPCR assay using clinical isolates from patients to screen for antibiotic resistance genes, a rapid molecular diagnostic method that is useful for the simultaneous detection of antibiotic resistance genes and allows for their early detection with sensitivity and specificity [40].

## 2. Materials and Methods

### 2.1. Clinical Isolates and Reference Strains

The 172 clinical isolates identified by results of the AST included in the study were 52 MRSA isolates and 39 MSSA isolates, collected during the period from August 2015 to June 2018, obtained from blood cultures and 80 VRE isolates and one vancomycin-susceptible enterococci (VSE) isolate, collected during the period January 2022 to December 2022, from the stool and urine cultures of patients. This study was approved by the Institutional Review Board of the Catholic University of Pusan (approval no. CUPIRB-24-01-008). Reference strains were obtained from American Type Culture Collection (ATCC; Manassas, VA, USA), National Culture Collection for Pathogens (NCCP; Cheongju, Chungcheongbuk-do, Republic of Korea), and Korean Collection for Type Cultures (KCTC; Taejon, Republic of Korea; Appendix A). All clinical isolates and bacterial reference strains were routinely grown on brain heart infusion (Difco, Le Pont-de-Claix, France) overnight at 37 °C.

### 2.2. Antimicrobial Susceptibility Testing (AST)

The AST of the clinical isolates was performed using BACTEC FX (Becton Dickinson, Sparks, MD, USA) and interpreted according to the CLSI guidelines [14]. The sample solution was simultaneously inoculated on a blood agar plate (BAP) and MacConkey agar plate for the isolation and incubation of clinical isolates, and cultured in a 35 °C, 5% CO_2_ incubator for 24 h.

### 2.3. Genomic DNA Extraction

Genomic DNA from reference strains and clinical isolates was extracted using the boiling method with Chelex-100 Resin (Bio-Rad Laboratories, Hercules, CA, USA) [41,42]. The clinical isolates from patients are routinely cultured on selective agar plates to distinguish between MRSA and VRE. Pure colonies were picked resuspended in BHI broth for the extraction of genomic DNA. Then, 1 mL of the culture broth of the reference strains and isolates were centrifuged at 13,000 rpm for 1 min, then frozen in a deep freezer for 10 min; they were mixed with 200 μL of 5% Chelex-100 solution and boiled at 100 °C for 10 min. The 5% Chelex-100 solution contained 5 g of Chelex-100 resin dissolved in 100 mL TE buffer. The samples were then centrifuged at 3000 rpm for 10 min. Only the upper soluble fraction containing genomic DNA was collected and transferred to a 1.5 mL sterilized microcentrifuge tube and stored at −20 °C. The DNA concentration and quality was estimated using a NanoDrop™ 2000 spectrophotometer (Thermo Fisher Scientific, Waltham, MA, USA).

### 2.4. Development of Primers and Hydrolysis Probes for mecA and vanA-vanB Antibotic Resistance Genes

To detect resistance genes, sequences of *mecA*, *vanA*, and *vanB* were collected from GenBank (www.ncbi.nlm.nih.gov/genbank/ (accessed on 29 August 2024); Appendix A). Real-time PCR primers and fluorogenic probes targeting *mecA*, *vanA*, and *vanB* as resistance genes were designed to distinguish them from other bacterial, fungal, and human genomic DNA. The primers amplified approximately 184 and 206 bp of the *mecA-* and *vanA*-*vanB*-specific fragment of resistance genes, respectively. For the one-step multiplex qPCR assay, specific hydrolysis probes for *mecA* and *vanA*-*vanB* genes were designed and labeled with different hydrolysis dyes (Table 1 and Figure 1). To design primer–probes in specific regions for the resistance gene *mecA* of MRSA and *vanA*-*vanB* of VRE, a multiple sequence alignment was performed using Clustal Omega Tools (https://www.ebi.ac.uk/jdispatcher/msa/clustalo; accessed on 20 June 2024; EMBL-EBI, Hinxton, UK) for nucleotide sequences with accession numbers of target genes from the National Center for Biotechnology Information (Appendix A). Then, the primer and probe sequences were selected from regions without mismatches, and the secondary structures of the primer and probe, and the possibility of dimer formation and hairpins were evaluated using IDT Oligo analyzer Tools (https://sg.idtdna.com/calc/analyzer; accessed on 20 June 2024; Integrated DNA Technologies, Coralville, IA, USA).

### 2.5. One-Step Multiplex qPCR Assay

The one-step multiplex qPCR assay was performed using a CFX-96 real-time PCR system (Bio-Rad, Hercules, CA, USA). Amplification was carried out in 20 μL reaction volumes, including 10 μL of 2× Thunderbird probe quantitative PCR mixture (Toyobo, Osaka, Japan), 1 µL of each 10 µM forward primer and reverse primer for *mecA*, 1 µL of 1 µM hydrolysis probe for *mecA*, 1 µL of each 10 µM forward primer and reverse primer for *vanA-vanB*, 1 µL of 1 µM hydrolysis probe for *vanA*-*vanB*, 2 µL template DNA, and distilled water. The conditions for the amplification were 95 °C for 10 min, followed by 40 cycles of 95 °C for 15 s, 55 °C for 30 s, and 72 °C for 30 s. Each sample was amplified in a single reaction containing primers and probes specific for two targets, including the *mecA* gene of MRSA and *vanA*-*vanB* gene of VRE. We used different fluorescent dyes as individual targets (Table 1). These probes, with unique emission spectra as fluorescent dyes, can be added to the same PCR amplification reaction for the simultaneous detection of multiple targets in a single assay. Multiplexing tends to significantly hamper the amplification performance for many combinations; therefore, the reagent mixtures and qPCR conditions were carefully evaluated. We set the optimal conditions under which the Ct value and amplification curve of the multiplex were improved compared to those when detection was performed in a separate reaction. Therefore, the qPCR method was developed using the primers and hydrolysis probes designed in this study and was optimized for amplifying two target genes with each fluorescent probe in one tube by adjusting the qPCR cycle conditions (Figure 2).

### 2.6. Analytical Sensitivity and Specificity

To determine the analytical sensitivity for the detection of *mecA* and *vanA*-*vanB* genes, 10-fold serially diluted synthetic plasmid DNA containing the *mecA* or *vanA*-*vanB* sequence, ranging from 1.0 × 10^8^ to 1.0 × 10^0^ copies/μL, was analyzed. Plasmid DNA genes were synthesized at the Macrogen Institute (Seoul, Republic of Korea), and each sample for the one-step multiplex qPCR assay was analyzed in triplicate. The synthetic plasmid DNA was quantified using the NanoDrop™ 2000 spectrophotometer (Thermo Fisher Scientific) to determine the DNA concentration. The DNA Copy Number and Dilution Calculator web tool (https://www.thermofisher.com/kr/ko/home/brands/thermo-scientific/molecular-biology/molecular-biology-learning-center/molecular-biology-resource-library/thermo-scientific-web-tools/dna-copy-number-calculator.html; accessed on 27 June 2024) was used to calculate the recombinant DNA copy number to dilute the first dilution with a known number of DNA copies. Synthetic plasmid DNA was used to determine specificity for the detection of the *mecA* and *vanA*-*vanB* genes. Cross-reactivity tests were conducted on target and nontarget substances to confirm that no other substances were detected by the probes in the sample. For the one-step multiplex qPCR assay, each sample was analyzed in triplicate.

### 2.7. Sequencing Analysis

To perform the bacterial 16S rRNA-based molecular analysis, PCR products amplified from clinical isolates were sequenced using *mecA* and *vanA*-*vanB*, and universal primers: F27 (5′-AGAGTTTGATCMTGGCTCAG-3′) and R1492 (5′-TACGGYTACCTTGTTACGACTT-3′) [43]. Sequences were analyzed at the Bionics Institute using Sanger sequencing in an Applied Biosystems 3730XL DNA Analyzer (Thermo Fisher Scientific).

### 2.8. Statistical Analysis

To analyze the qPCR data, statistical analyses were conducted using Prism 8 (GraphPad Software, San Diego, CA, USA). Receiver operating characteristic (ROC) curves and areas under the curves (AUCs) were used to analyze the DNA of clinical isolates, MRSA, and methicillin-susceptible *S. aureus* (MSSA), which have the potential to be used as samples for the diagnosis of resistant infections. A *p*-value < 0.05 was considered statistically significant.

## 3. Results

### 3.1. Antimicrobial Susceptibility Testing (AST) Data of Clinical Isolates

AST result data for 172 clinical isolates were provided, and according to the AST profile results, 30.2% (52/172) were confirmed to be resistant to oxacillin, 22.7% (39/172) were confirmed to be susceptible to oxacillin, 46.5% (80/172) were confirmed to be resistant to vancomycin, and 0.6% (1/172) were confirmed to be susceptible to vancomycin (Table 2). The Korea Centers for Disease Control and Prevention (KCDC) operates infectious disease surveillance systems to monitor national disease incidence [44]. Therefore, because VSE isolates are difficult to collect as clinical samples, it was difficult to include additional cases other than the one case included in this study.

### 3.2. Analytical Sensitivity and Specificity of the One-Step Multiplex qPCR Assay

To confirm the analytical sensitivity and specificity of primers and probes designed in this study, synthetic plasmid DNA was used. The synthetic plasmid gene size of *mecA* and *vanA*-*vanB* were 224 bp and 246 bp, respectively (Appendix A). This one-step multiplex qPCR assay amplified the *mecA* and *vanA-vanB* target genes from each hydrolysis probe, respectively, and no cross-reactivity was found (Figure 3A). Primers and hydrolysis probes were synthesized at the Macrogen Institute (Seoul, Republic of Korea). The linear standard curve of the real-time qPCR for the *mecA* and *vanA-vanB* genes is shown in Figure 3B,C, respectively. The calculation of the E and R^2^ values for *mecA* and *vanA*-*vanB* genes was performed using each pair of primers and probe. The qPCR results showed that the R^2^ values for the synthetic plasmid DNA *mecA* and *vanA*-*vanB* were 0.998 and 0.999, respectively, indicating good linearity. The amplification efficiencies for the plasmid of *mecA* and *vanA*-*vanB* genes was at 94.9% and 100.1%, respectively. When the Ct value was verified using the reference strains as positive controls, Ct values of >30 were considered inadequate. In this study, the negative cut-off value was set to ≥30. Therefore, the analytical sensitivity for the target of *mecA* and *vanA*-*vanB* genes were confirmed to be 7.94 × 10^2^ copies/μL and 1.10 × 10^3^ copies/μL, respectively (Table 3). The plasmid genes synthesized as positive controls were confirmed by the one-step multiplex qPCR assay and by sequencing (Appendix A). Additionally, this one-step multiplex qPCR assay amplified the three reference strains of the positive controls but did not amplify the two reference strains of the negative controls, and no cross-reactivity was found (Appendix A).

### 3.3. Identification of MRSA, MSSA, VRE, and VSE Isolates Using One-Step Multiplex qPCR Assay

To further validate this assay for detecting *mecA*, *vanA*, and *vanB* genes, the DNA extracted from clinical isolates was used as the template DNA. The detection of the target gene for clinical isolates was confirmed using the one-step multiplex qPCR assay, and the results of MRSA, MSSA, VRE, and VSE isolates were 50 (96.2%), 39 (100%), 80 (100%), and 1 (100%), respectively, and were similar to the microbiological culture and AST results. The clinical isolates of MRSA and VRE were confirmed using the *mecA* primer, and *vanA*-*vanB* primers were used for amplifying the DNA between positions 27 and 1492 of bacterial 16S rRNA. Universal primers (27F and 1492R) are widely used for bacterial identification by analyzing the 16S rRNA gene [45,46] (Table 4). The sensitivity and specificity of the *mecA* gene for clinical *S. aureus* isolates, including MRSA and MSSA, were 97.44% (95% CI, 86.82–99.87%) and 96.15% (95% CI, 87.02–99.32%), respectively (Figure 4). The ROC AUC for the diagnosis of MRSA was 0.9798 (*** *p* < 0.0001).

### 3.4. Comparison of AST and One-Step Multiplex qPCR Assay for Clinical Isolates

Through AST, among 172 clinical isolates, 30.2% (52/172) were identified as MRSA, 22.7% (39/172) as MSSA, 46.5% (80/172) as VRE, and 0.6% (1/172) as VSE. Meanwhile, 29.1% (50/172) were confirmed to be *mecA* positive, 22.7% (39/172) were confirmed to be *mecA* negative, 46.5% (80/172) were confirmed to be *vanA*-*vanB* positive, and 0.6% (1/172) were confirmed to be *vanA-vanB* positive by the one-step multiplex qPCR assay (Figure 5). Therefore, overall, the AST results and *mecA* and *vanA*-*vanB* one-step multiplex qPCR analysis for clinical isolates tended to be 98.3% (169/172) consistent, except for three samples.

## 4. Discussion

Antimicrobial resistance is becoming increasingly serious owing to the use and abuse of a wide range of antibiotics, and the emergence of antibiotic-resistant bacteria poses a major threat to public health [47,48]. The rate of antibiotic resistance remains high; therefore, systematic management is required. In South Korea, the proportion of MRSA and VRE is still high, and the multi-drug resistance (MDR) rate of *S. aureus* and *E. faecium*, which are resistant to three or more classes of antibiotics, is also high at 46.9% and 57.9%, respectively [39]. Currently, genotypic molecular methods are used as gold standard diagnostic methods to confirm each antibiotic resistance target gene [49,50,51]. Therefore, this study established an approach showing an improved, simultaneous diagnostic detection over the existing genotypic molecular methods that was developed by enabling the simultaneous diagnostic detection of *mecA* and *vanA*-*vanB* genes. The worldwide incidence of MRSA was reported to be 2.3–69.1% [52] and the global pooled prevalence of MRSA was reported to be 14.69% [48]. According to the European Centre for Disease and Control 2018 report, the population-weighted prevalence of VRE across European countries was 17.3% in 2018 [53]. The overall prevalence of VRE in clinically isolated enterococci in Latin America has been reported to be 31% [53]. In South Korea, MRSA accounts for 60–70% of all isolates from tertiary hospitals [54]. In 2020, the incidence of MRSA in general and nursing hospitals in South Korea was reported to be 47.4% and 88.8%, respectively [55]. The vancomycin resistance rate of *E. faecium* in 2020 was 38.6% in general hospitals and 94.9% in nursing hospitals in South Korea [55]. Six types of antimicrobial resistant bacteria have been legally declared as the causative agents behind infectious diseases in South Korea and managed since the Infectious Disease Control and Prevention Act was enacted in 2009 [56]. The six types of antibiotic-resistant bacteria in South Korea include MRSA, VRE, VRSA, MRPA, MRAB, and CRE [5,6,10,56]. After the start of the Korea Global AMR Surveillance System (Kor-GLASS) in 2016 in South Korea, methicillin resistance rates of MRSA and vancomycin resistance rates of VRE were reported to be 45.2% and 37.7% in 2021, respectively [57,58]. In clinical laboratories, antibiotic resistance tests are performed to identify antibiotic-resistant bacteria, with culture-based screening being the most conventional gold standard method [49,59,60]. General methods for diagnosing antibiotic-resistant bacteria include phenotypic AST and genotypic molecular methods [49,50,51]. The classical and gold standard phenotypic methods of AST for *S. aureus* include the broth microdilution and disk diffusion methods [60]. Additionally, traditional methods for detecting VRE identify bacteria through Gram staining, standard biochemical tests, and characterize antimicrobial resistance through disc diffusion, minimum inhibitory concentration, and broth dilution [35,61,62]. Culture-based screening involves the use of a selective medium specific for the target bacteria, culturing it for at least one day, and then identifying the bacteria growing on the medium [12,59,63], which ultimately takes at least 48–72 h [12,64,65,66,67]. However, culture-based methods are still used despite the disadvantage of taking a long time [47,68,69,70].

Genetic methods to detect MRSA and VRE have also recently been used. Genotypic molecular methods include PCR and whole genome sequencing and are performed by targeting resistant genes [19,20,50,52,71]. The molecular methods involved the detection of the methicillin resistance gene (*mecA*) and vancomycin resistance genes (*vanA* and *vanB*) [21,47,72,73]. Molecular-based diagnostic methods have shortened the time required to detect and identify antibiotic-resistant infections [74]. Therefore, it is important to develop a diagnostic method that can efficiently and effectively detect the antibiotic-resistant bacteria that frequently occur in the clinical environment. In this study, clinical isolates were identified as 52 MRSA, 39 MSSA, 80 VRE, and 1 VSE isolates according to AST (Table 2). According to the one-step multiplex qPCR results, 50 of 52 MRSA isolates were confirmed *mecA* positive, 39 of 39 MSSA isolates were *mecA* negative, 80 of 80 VRE isolates were positive for *vanA-vanB*, and 1 of 1 VSE isolate was *vanA-vanB* positive (Table 4). Therefore, the AST results and one-step multiplex real-time PCR results for clinical isolates were 97.8% (89/91) consistent for 91 *S. aureus* isolates and 98.8% (80/81) for 81 enterococci isolates (Figure 5).

Recently, molecular diagnostic methods have been used to detect and diagnose infections by targeting the *mecA* gene in MRSA, and the *vanA* and *vanB* genes in VRE [21,40]. Molecular AST methods are based on the direct detection of the antimicrobial resistance genes in bacterial isolates [12,75]. The *mecA* gene is known to be 2.1 kb in length. This gene is carried by a *mec* element or staphylococcal cassette chromosome *mec* (SCC*mec*) and inserted near the origin of chromosomal replication. Several types of glycopeptide resistance are known in enterococci, and the various types can be genotyped based on sequence differences in the ligase genes. The *vanA* gene cluster is located on the non-conjugative transposon Tn*1546*, which may be part of a mobile chromosome or an extrachromosomal element, while the *vanB* gene cluster is more heterogeneous at the DNA sequence level and the *vanB2*-operon appears to be linked to the putative conjugative transposon Tn*5382*/Tn*1549* [76]. Compared to traditional PCR analysis, real-time PCR can be processed quickly from sample preparation to results in less than 5 h and provides improved sensitivity and specificity. As real-time PCR is performed in a closed system, the potential for contamination is significantly reduced, and overall costs have decreased as the number of commercially available real-time PCR chemicals and platforms increase [38]. Moreover, for more rapid diagnosis from the existing PCR method, a molecular method that improves time and cost is needed. Accordingly, introducing simultaneous diagnostic detection methods for MRSA and VRE can reduce time and increase specimen throughput [38]. In addition, MRSA and VRE account for most of the antibiotic resistance in South Korea, and because they have a high multidrug resistance rate [39], detection efficiency can be increased by using the one-step multiplex qPCR assay. The one-step multiplex qPCR assay can simultaneously detect each target gene (*mecA* and *vanA-vanB*) with different fluorescence compared to the single real-time PCR detection method, which detects only a single target. Accordingly, this one-step multiplex qPCR assay for detecting MRSA and VRE-resistant bacteria uses different fluorescent dyes in the primer and probe design process. This one-step multiplex qPCR assay has clinical utility because it increases specimen throughput and saves time and cost to identify each target [38,77,78,79].

Currently, commercially available kits based on molecular genetic methods target *mecA* genes for MRSA: the GeneOhm™ MRSA assay (BD Diagnostics, San Diego, CA, USA) [80] and Xpert MRSA kit (Cepheid, Sunnyvale, CA, USA) [81]. Additionally, the targets for VRE are the *vanA* or *vanB* genes using the Anyplex^TM^ VanR Real-time Detection kit (VanR) (Seegene, Inc., Seoul, Republic of Korea) [82], and the iNtRON VRE *vanA*/*vanB* real-time PCR detection kit (iNtRON Biotechnology, Seongnam, Republic of Korea) [83]. The sensitivity and specificity for MRSA detection were 92.5% and 96.3% for the GeneOhm MRSA assay [80] and 89.5% and 97.3% for the Xpert MRSA assay [81], respectively. In the study, the sensitivity and specificity of qPCR for the diagnosis of MRSA were 97.44% (95% CI, 86.82–99.87%) and 96.15% (95% CI, 87.02–99.32%; Figure 4), which were similar or significantly higher than those in the previous study [80,81]. The sensitivity and specificity for VRE detection were 86.0% and 83.3%, respectively, for the iNtRON *vanA*/*vanB* assay [83]. The *vanA*-*vanB* primer–hydrolysis probe set developed for this study was designed for the detection of the *vanA*-*vanB* gene. In South Korea, because the Korea Centers for Disease Control and Prevention (KCDC) operates an infectious disease surveillance system to monitor nationwide disease incidence [44], VSE is difficult to collect as clinical samples, and only one clinical VSE isolate was included in this study. Another GeneSoC^®^ system kit that targets the *mecA* gene had an analytical sensitivity of 10 copies/μL, and the detection limit using MRSA bacterial burden is reported to be 10^3^ CFU/mL [84]. Similarly, in the qPCR method developed in this study, the analytical sensitivity was 7.94 × 10^2^ and 1.10 × 10^3^ copies/μL targeting *mecA* and *vanA*-*vanB* genes, respectively (Figure 3 and Table 3). However, these available diagnostic kits only detect a single resistance gene of a single resistant bacterium, making it difficult to simultaneously diagnose MRSA and VRE, which are the most common bacteria in South Korea. We aimed to develop a new diagnostic method that can simultaneously detect multiple resistance genes of MRSA and VRE in a single tube and one reaction of the one-step multiplex qPCR assay. Accordingly, our research team developed a molecular diagnostic method to simultaneously detect two types of antibiotic genes, *mecA* and *vanA*-*vanB* genes, to identify antibiotic-resistant bacteria, such as MRSA and VRE. Subsequently, during the development of this study, this one-step multiplex qPCR assay was able to detect multiple target genes using different fluorescent dyes, thus reducing the time, costs, and labor required to diagnose resistant bacterial infections compared to that of other detection methods that only detect a single target gene [77]. Considering the current high incidence of MRSA and VRE among the six resistant bacteria in South Korea, the simultaneous detection of *mecA* and *vanA*-*vanB* genes using this newly developed one-step multiplex qPCR assay is a very useful method for accurately identifying resistant bacteria. Additionally, when mass testing becomes possible at a price similar to that of conventional culture-based methods, it is expected that the number of patients for antibiotic resistance screening tests will expand from specific patients to all hospitalized patients. Therefore, the molecular diagnostic method using this newly developed one-step multiplex qPCR assay can provide accurate and rapid results for the treatment of patients with MRSA and VRE infections. Because this study focused on simultaneously detecting the *mecA* gene and *vanA*-*vanB* gene using different fluorescent probes, the *mecA* gene possessed by both MRSA and methicillin-resistant coagulase negative staphylococci (MR-CoNS) could be detected, but the limitation was that it could not distinguish between MRSA and MR-CoNS. In the future, the researchers plan to collect MR-CoNS samples to further study the differences between MRSA and MR-CoNS. In addition, among *Enterococcus*, *vanA*-*vanB* resistance genes were be detected for *E. faecalis* and *E. faecium*, but species discrimination was not possible. In future studies, more clinical specimens will be collected, and the effectiveness of the *mecA* and *vanA*-*vanB* gene biomarkers developed in this study should be verified through statistical analysis after the additional collection of clinical isolates of MSSA and VSE, as well as MRSA and VRE. As a routine test, culture methods are time-consuming and produce negative results in most cases, leading to the widespread use of antibacterial agents. Although molecular diagnostic research is underway to directly detect pathogens in blood, there are still many difficulties in applying this to routine use [74,85,86]. The molecular diagnostic platform currently offered by DreamDX demonstrates the potential for the simultaneous diagnosis of MRSA and VRE. Therefore, additional tests are needed after collecting samples directly from patients. Performing molecular testing directly on blood without culture has the potential to provide timely and appropriate treatment to patients.

## 5. Patents

A patent is in process for the primer–probe set for detecting methicillin- and vancomycin-type antibiotic resistance genes: Composition for detecting antibiotic-resistant bacteria and detection method using the same (Patent No. 10-2023-0097961).

## Figures and Tables

**Figure 1 pathogens-13-00853-f001:**
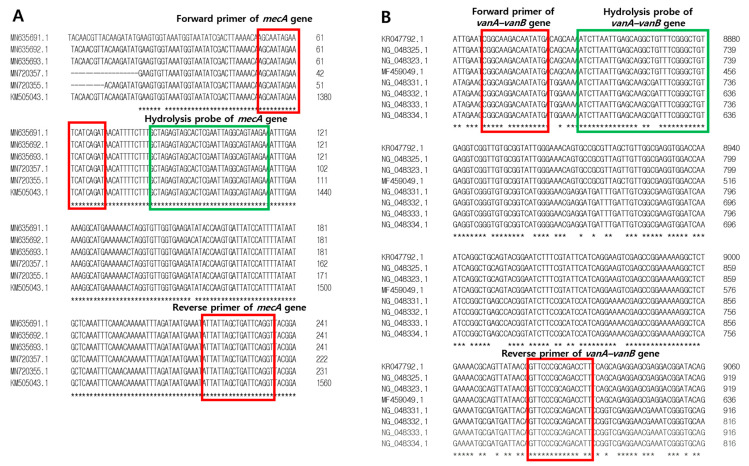
An alignment (5′–3′) of the primers and probe with (**A**) the *mecA* gene and (**B**) the *vanA-vanB* gene. Forward and reverse primer sequences are shown in red frames and probe sequences are shown in green frames. The “*” character means that the residue or nucleotide is identically conserved.

**Figure 2 pathogens-13-00853-f002:**
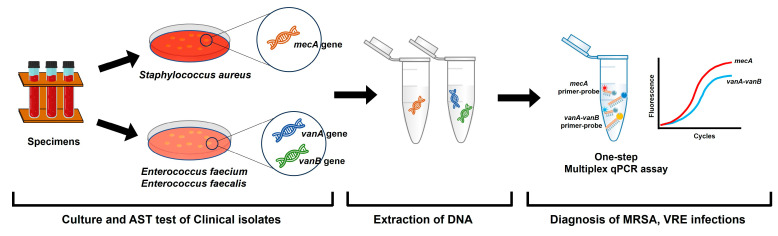
Flow chart for sample processing by the one-step multiplex qPCR assay protocol for MRSA and VRE detection.

**Figure 3 pathogens-13-00853-f003:**
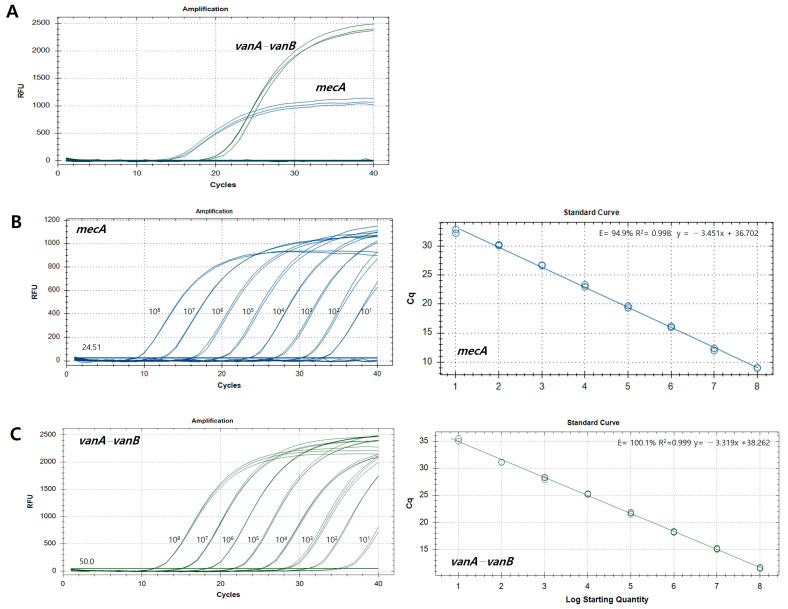
Results of specificity and analytic sensitivity using synthetic plasmid DNA. (**A**) Amplification curve of the *mecA* and *vanA-vanB* genes. (**B**) Amplification curves and standard curves of log10 genome versus Ct value of *mecA* genes. (**C**) Amplification curves and standard curves of log10 genome versus Ct value of *vanA-vanB* genes.

**Figure 4 pathogens-13-00853-f004:**
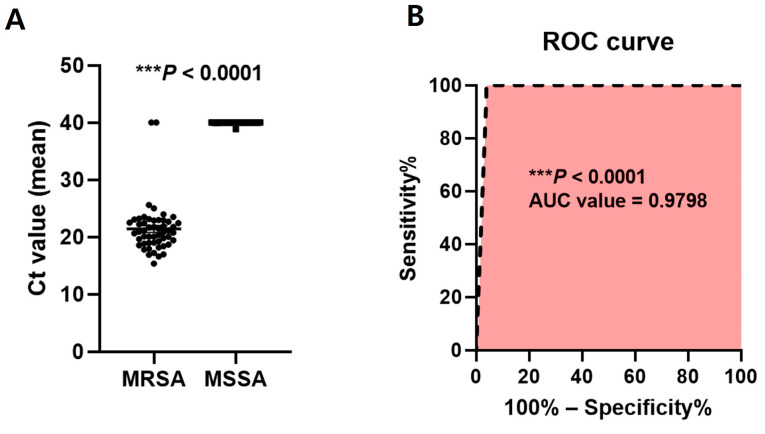
Results of specificity and sensitivity using the *mecA* primer–hydrolysis probe set. (**A**) The Ct value (mean) of *mecA* gene among the MRSA and MSSA samples. (**B**) Receiver operating characteristic (ROC) curve analysis to evaluate the diagnostic value of the *mecA* primer–hydrolysis probe set. Data are presented as mean ± SEM (standard error of the mean). Abbreviations: MRSA, methicillin-resistant *S. aureus*; MSSA, methicillin-susceptible *S. aureus*; ROC, receiver operating characteristic.

**Figure 5 pathogens-13-00853-f005:**
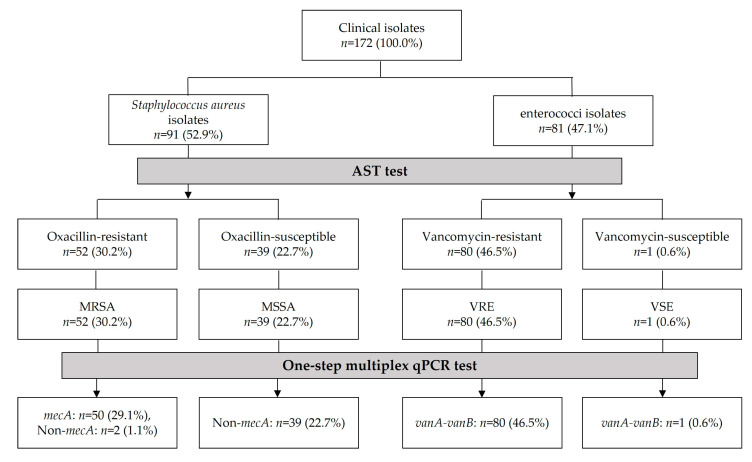
Comparison of the results of AST and multiplex qPCR in clinical isolates.

**Table 1 pathogens-13-00853-t001:** The design of primers and probe sequences for *mecA* and *vanA*-*vanB* genes on one-step multiplex qPCR assay.

Target	Primers and Probes	Sequences (5′-3′)	Amplicon Size (bp)	Length (mer)
*mecA*	Forward primer	AGCAATAGAATCATCAGAT	184	19
	Reverse primer	ACCTGAATCAGCTAATAAT		19
	Probe	FAM-GCTAGAGTAGCACTCGAATTAGGCAGTAAGA-TAMRA		31
*vanA-vanB*	Forward primer	CGGCAGGACAATATGA	206	16
	Reverse primer	AATGTCTGCGGGAAC		15
	Probe	HEX-ATCTTAATTGAGCAAGCGATTTCGGGCTGT-TAMRA		30

**Table 2 pathogens-13-00853-t002:** Antimicrobial susceptibility testing results of clinical isolates.

Antimicrobial Agent	Standard Interpretation (CLSI) [14]			Total *n* (%)
	Resistant *n* (%)	Intermediate *n* (%)	Susceptible *n* (%)	
Oxacillin	52 (30.2%)	0 (0%)	39 (22.7%)	172 (100%)
Vancomycin	80 (46.5%)	0 (0%)	1 (0.6%)	

Oxacillin is a narrow-spectrum beta-lactam antibiotic of the penicillin class developed by Beecham, and vancomycin is a glycopeptide antibiotic medication used to treat a number of bacterial infections.

**Table 3 pathogens-13-00853-t003:** Mean Ct values for qPCR carried out on serial dilutions of plasmid DNA of *mecA* and *vanA-vanB* genes.

Plasmid DNA of Resistance Gene	Plasmid DNA Content	Mean Ct Value	SD	Log Value	Copies per Reaction (copies/μL)
*mecA* gene	2.58 ng	9.04	0.00	8.02	1.04 × 10^8^
	258 pg	12.18	0.18	7.11	1.28 × 10^7^
	25.8 pg	16.09	0.11	5.97	9.42 × 10^5^
	2.58 pg	19.53	0.17	4.98	9.49 × 10^4^
	258 fg	23.15	0.18	3.93	8.48 × 10^3^
	25.8 fg	26.70	0.00	2.90	7.94 × 10^2^
	2.58 fg	30.17	0.09	1.89	7.84 × 10^1^
	258 ag	32.54	0.32	1.21	1.61 × 10^1^
*vanA*-*vanB* gene	964 pg	11.61	0.11	8.03	1.07 × 10^8^
	96.4 pg	15.15	0.07	6.96	9.22 × 10^6^
	9.64 pg	18.26	0.07	6.03	1.07 × 10^6^
	964 fg	21.72	0.14	4.99	9.66 × 10^4^
	96.4 fg	25.25	0.05	3.92	8.31 × 10^3^
	9.64 fg	28.17	0.21	3.04	1.10 × 10^3^
	964 ag	31.14	0.01	2.15	1.40 × 10^2^
	96.4 ag	35.26	0.24	0.9	0.83 × 10^0^

The copies per reaction of *mecA* gene were 2.58 ng = 1.04 × 10^8^ copies, 258 pg = 1.28 × 10^7^ copies, 25.8 pg = 9.42 × 10^5^ copies, 2.58 pg = 9.49 × 10^4^ copies, 258 fg = 8.48 × 10^3^ copies, 25.8 fg = 7.94 × 10^2^ copies, 2.58 fg = 7.84 × 10^1^ copies, and 258 ag = 1.61 × 10^1^ copies. The copies per reaction of *vanA*-*vanB* gene were 964 pg = 1.07 × 10^8^ copies, 96.4 pg = 9.22 × 10^6^ copies, 9.64 pg = 1.07 × 10^6^ copies, 964 fg = 9.66 × 10^4^ copies, 96.4 fg = 8.31 × 10^3^ copies, 9.64 fg = 1.10 × 10^3^ copies, 964 ag = 1.40 × 10^2^ copies, and 96.4 ag = 0.83 × 10^0^ copies. Abbreviations: SD, standard deviation.

**Table 4 pathogens-13-00853-t004:** Sample types and qPCR results of clinical isolates.

Clinical Isolates	Mutiplex qPCR Using *mecA* and *vanA*-*vanB* Primers-Hydrolysis Probes				Sequencing Results Using *mecA* Primer, *vanA*-*vanB* Primer, and Universal Primer		
	*mecA* Positive Ct Mean ± SD (*n*, %)	*mecA* Negative Ct Mean ± SD (*n*, %)	*vanA*-*vanB* Positive Ct Mean ± SD (*n*, %)	*vanA*-*vanB* Negative Ct Mean ± SD (*n*, %)	*mecA* Positive Using *mecA* Primer (*n*, %)	*van* Positive Using *vanA-vanB* Primer (*n*, %)	Using Universal Primer (*n*, %)
MRSA isolates (*n* = 52, 100%)	20.7 ± 2.3 (50, 96.2%)	40.0 ± 0.0 (2, 3.8%)	(0, 0%)	40.0 ± 0.0 (52, 100%)	(50, 96.2%)	-	(52, 100%)
MSSA isolates (*n* = 39, 100%)	(0, 0%)	40.0 ± 0.2 (39, 100%)	(0, 0%)	40.0 ± 0.0 (39, 100%)	-	-	-
VRE isolates (*n* = 80, 100%)	(0, 0%)	39.8 ± 1.2 (80, 100%)	18.3 ± 1.4 (80, 100%)	(0, 0%)	-	(80, 100%)	(80, 100%)
VSE isolate (*n* = 1, 100%)	(0, 0%)	40.0 ± 0.0 (1, 100%)	16.8 ± 0.0 (1, 100%)	(0, 0%)	-	(1, 100%)	(1, 100%)
Total isolates (*n* = 172, 100%)	20.7 ± 2.3 (50, 29.1%)	39.9 ± 1.0 (122, 70.9%)	18.3 ± 1.4 (81, 47.1%)	40.0 ± 0.0 (91, 52.9%)	-	-	-

Abbreviations: SD, standard deviation; MRSA, methicillin-resistant *Staphylococcus aureus*; MSSA, methicillin-susceptible *S. aureus*; VRE, vancomycin-resistant enterococci; VSE, vancomycin-susceptible enterococci.

## Data Availability

The original contributions presented in the study are included in the article/Appendix A; further inquiries can be directed to the corresponding authors.

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
