# Peer review of "Development of a One-Step Multiplex qPCR Assay for Detection of Methicillin and Vancomycin Drug Resistance Genes in Antibiotic-Resistant Bacteria"

_pathogens, 2024, doi:10.3390/pathogens13100853_

Round 1
Reviewer 1 Report
Comments and Suggestions for Authors
The manuscript “Development of a One-Step Multiplex qPCR Assay for Detection of Methicillin and Vancomycin Drug Resistance Genes in Antibiotic Resistant Bacteria” by J. Lee and coauthors develops a multiplex qPCR method for the detection of methicillin-resistant Staphylococcus aureus (MRSA) and vancomycin-resistant enterococci (VRE). They use reference strains, synthetic plasmids and clinical isolates to evaluate their method and compare it with the results of the classic antimicrobial susceptibility testing (AST). The sensitivity and analytical specificity of the method present sufficient values ​​to consider the reproducibility and application of it.
In general terms, the method is useful and valid, as it speeds up the detection of MRSA and VRE in clinical cases. Methodologically, it presents the standard procedures to develop and validate a qPCR kit. However, there are some limitations, notably the absence of VSE isolates and the asynchrony between clinical isolates that distort some analyses. In addition, there are some errors in writing, contextualization, and omission of information. These points will be observed section by section below.
Abstract
Line 26: add the term "multiplex" before "...real-time polymerase chain reaction...")
Introduction
Line 50: Correct the mistyping error "bcateria"
Line 53: Be more precise and detail the AST, which consists of culture in medium with an antimicrobial agent where growth inhibition is evaluated.
Line 54: Authors must properly cite here the updated version of the CLSI standards (as cited in 2.2., ref 22).
Line 63: After this paragraph, it is necessary to give a genetic context about the mecA, vanA, vanB genes. Where they are found (chromosome, plasmid,...) and how they are transmitted. This is important to understand the number/proportion of gene sequences detected by each bacteria.
The authors need to provide clarifying context of the genetic similarities and differences between VRE and VRSA. Does the first include the second? This is important because they use the sequence of a VRSA in the synthesis of the synthetic control of vanA and vanB.
Lines 84-89: According to the procedures and methods used in this manuscript, this multiplex qPCR assay was validated in “clinical isolates.” Specify and define the limitations of use with respect to the source (pure colonies, liquid culture, original clinical sample, ...).
Methods
Line 105: What is the point of inoculating the microbial culture on MacConkey agar plate? Justify.
Section 2.2.: If the authors highlight that the importance of using a molecular kit is the time gain compared to classical AST methods, why did they not implement a Fast qPCR reaction?
Line 132: The final concentration of primers and probes used in qPCR assays is usually much lower than 10 uM each. Verify.
Line 135: Strictly, the reaction is designed to detect TWO targets (mecA and vanA-vanB) since it does not differentiate vanA and vanB. Clarification is necessary throughout the manuscript.
Section 2.7.: Indicate which or how many bacteria this procedure was performed on. On the other hand, mention how the resulting sequences were assembled and analyzed.
Results
Section 3.1.: These results are intriguing. The authors indicate that all enterococci (n=81) analyzed have total or intermediate resistance to vancomycin. This is difficult to believe unless these bacteria have been previously analyzed and determined to be VRE, indicating the need to clarify it in section 2.1. In any case, it is a limitation not to use VSE clinical samples.
Section 3.2.: This section should be transferred to materials and methods. Lines 200-203 can be included in section 3.3.
Lines 215-216: Specify if the calculation of E and R2 was carried out with each pair of primers in a separate mix or the 4 primers together for each case. This may change the values.
Lines 218-219: Because the typically accepted efficiency values ​​range between 90% to 110%, the authors should mention it as a limitation or disadvantage in the discussions.
Figure S1: In the legend it is mentioned "did not amplify the 14 reference strains of the negative controls". Throughout the manuscript the authors indicate that they used only 2 negative reference controls. Verify.
Section 3.5.: This paragraph needs to be improved. Sequencing results were 96.2% and 100% of what?. The term "similar" is purposely used and comparison of exact values ​​is omitted. The universal reaction for the 16S gene uses 2 primers (27F and 1492R) and not the "universal primer 27F1492R".
Figure 5: According to table 2, there are no susceptible vancomycin but there is an intermediate.
Discussion:
There are some redundant ideas in this section (gold standard AST methods, speed of qPCR compared to classical methods, simultaneous detection using multiplex qPCR, among others). I recommend reviewing and rephrasing the discussion.
It is necessary to include discussions about the genetic context of the mecA, vanA and vanB genes in relation to their transmission, number of copies per bacteria and their relationship with CFU, possible defective genes due to mutation or storage conditions.
Lines 282-284: These analyzes are inappropriate in the context of comparative incidences as indicated in the previous lines. The samples used in the present study have different origins and times (according to what was reported in sections 2.1. and 3.1.) so they cannot be grouped to give relative prevalences.
Lines 308-311: According to the methods of this study it is understood that the bacteria were isolated and therefore cultured before DNA extraction. This discussion utopianly refers to direct testing without bacterial culture. Rephrase.
Line 359: It is confusing and wrong to alternate terms of copies/μL and CFU/mL. The authors performed serial dilutions based on copy counts of synthetic gene fragments which are not necessarily equivalent to one CFU. To make this statement they need to start from serial dilutions of CFU.
Author Response
Dear Reviewer
Thank you very much for taking the time to review this manuscript.
Please find the detailed responses below and the corresponding revisions in the re-submitted files.
Please see the attachment.
Kind Regards

Reviewer 2 Report
Comments and Suggestions for Authors
This study is interesting and also practically useful for clinical microbiology laboratory. Presentation in the manuscript is generally well, but this reviewer suggest following points to revise the manuscript.
1. Similar methods of qPCR to detect mecA in S. aureus and vanA/B gene in Enterococcus might have been published previously. What is the novel point of method in thie study? What were improved in this study? Such ponts should be more clearly written.
2. Figure 1. From "specimens" at left side, MRSA and VRE were shown. However, it is somewhat strange, because these isolates have not yet been confirmed as MRSA/VRE at this stage. These should be just "S. aureus" "Enterococcus", etc.
3. In this qPCR, species identification is not included. How did authors confirm bacterial species? Practically, if clinical specimens are examined, mecA is detected also from coagulase-negative staphylococcus. How are MRSA and MR-CoNS discriminated? Is it disregarded? For Enterococcus, are E. faecalis, E. faecium, and other species able to identified? Authors' idea regarding species identification must be described.
4. Table 5, rightmost colums, "universal primer" "100%) are not understandable. Authors used these primer directed to 16S rRNA. Readers probably think that these primers were used to identify bacterial species, by sequencing of PCR product. In this regard, check descripions in Methods and this Table, and correct appropriately.
Comments on the Quality of English LanguageGenerally OK.
Author Response

(The authors gave the same response as above.)

Round 2
Reviewer 2 Report
Comments and Suggestions for Authors
Revised manuscript is well written.